# The health costs of losing political representation: Evidence from U.S. Presidential Elections

Sris Chatterjee[1], Iftekhar Hasan[1,2,3], Stefano Manfredonia [1]*

**1** Gabelli School of Business, Fordham University, New York, New York, United States of America, **2** Bank of Finland, Helsinki, Finland, **3** University of Sydney Business School, Sydney, Australia

\* smanfredonia1@fordham.edu

## Abstract

We investigate whether a change in political leadership affects health outcomes. To do so, we exploit turnover elections that move partisan individuals into and out of alignment with the party of the President. We document that the lack of political alignment has a negative, immediate, and long-lasting effect on health. We do not find any evidence that our results can be explained by other confounding trends or by changes in economic outcomes or other economic policies. Further results suggest that political sentiments and social isolation are important potential mechanisms in this setting and that lack of political representation affects the mental health of individuals.

## 1 Introduction

Several papers document a worrying increase in political polarization in the United States during the last decades [1–4]. Hostility toward political opponents increased substantially; both Democrats and Republicans claim that the other faction generates "very unfavorable" feelings, such as frustration, fear, and anger (Pew Research Center, 2016). Spatial partisan sorting is creating homogeneous political communities [5–7], and political minorities within these communities show apprehension to disclose their political preferences and are more likely to be subject to discrimination and social isolation [8,9]. Also, divergent views toward political matters have been shown to affect expectations about future economic outcomes; individuals are indeed more optimistic when they are affiliated with the party that controls the government, and this tendency has significantly increased in the last decade [10,11].

The purpose of our paper is to investigate whether a shift in political leadership and the lack of political representation can affect health outcomes. Divergent political opinions can indeed increase stress, anxiety, and the feeling of social isolation with a negative effect on physical and mental health [12–14]. Also, partisan biased beliefs about future economic growth and economic policies benefiting the related partisan group of the population can potentially affect households' welfare, consumption, and investment patterns (e.g., [15–17]), leading effectively to greater economic prosperity that can have a positive effect on community well-being.

**Data availability statement:** The data used in this study are administrative records provided by the National Center for Health Statistics (NCHS). Due to legal restrictions and data use

agreements, the data cannot be shared publicly. Access to the data can be requested at: dvsdatarequests@cdc.gov The NETS database has been acquired. Access to the data can be requested by contacting dwalls2@earthlink.net.

**Funding:** The authors received no specific funding for this work.

**Competing interests:** The authors declare that they have no competing interests.

To advance our research questions, we identify communities' health using administrative data on the universe of death certificates in the United States. Since our data comprehensively covers the whole US population, we can derive general estimates for the overall health costs related to a change in political leadership. However, mortality is an extreme health outcome, and therefore, our estimates will underestimate the true effect of a change in political leadership on health. For this reason, we also evaluate individuals' general health status by using a large survey database and corroborate our results in a robustness check using this as an alternative outcome variable.

Since we can not directly link information from death certificates to other sources that provide information on person-specific political preferences, we follow for identification purposes recent empirical literature and use residential location to form groups of potentially like-minded individuals [11,18–20]. Therefore, our main empirical strategy makes use of turnover Presidential elections that change counties' political alignment with the party of the President, with a particular focus on the elections of President Barack Obama first and Donald Trump later.

Exploiting cross-county differences in political preferences and using a differences-in-differences approach, we document a worsening in the health outcomes in losing counties after turnover Presidential elections. More specifically, we estimate an average yearly increase of 7 deaths per 100,000 population. Considering that the estimated value of a life in the United States according to the Federal Emergency Management Agency (FEMA) is US $7.5 million and that the average county population is equal to 104,487, the effect is also economically meaningful.

We provide evidence that our results are unlikely to be explained by other confounding factors. To do so, we show graphically that the parallel trend assumption for the validity of our identification strategy is likely to hold. We find that indeed Democratic and Republican counties have similar mortality patterns before the 2008 and 2016 turnover Presidential elections and divergent patterns afterward. Event-study estimates provide further empirical support for this hypothesis and show that a statistically significant effect appears only immediately after the turnover elections and is persistent in the following years.

Even though the parallel trend assumption is likely to hold, another potential concern of our approach is that other confounding county characteristics can also affect our findings. Ideally, we would like to know the political preferences of each individual; however, because of privacy-related issues, the identity of an individual and his/her political preferences are not available in the database of death certificates. To deal with this problem and show that potentially confounding county characteristics do not affect our results, we conduct an additional test and exploit a unique characteristic of the Obama Presidential election, viz, the fact that most of the black voters (95 % of the black population) cast their ballot for the Democrat Barack Obama (Pew Research Center, 2009). Exploiting this peculiarity, we build a new database at the county-race-year level and compare black mortality rates with the mortality rates of other races (White, Hispanic, and Asiatic). This approach is particularly useful since it allows us to compare individuals living in the *same county at the same period*, and it helps to provide further evidence on the validity of our identification strategy. In line with our main finding, we document a decrease in black mortality rates after the election of Obama.

Since our inferences are based on two turnover Presidential elections, we provide external validity to our results by considering a long cross-sectional survey database starting in the year 1973. Importantly for our purpose, this database provides us with information on the health of the individuals, but also on their political preferences. Using this long survey database, we document that the correlation between political preferences and health depends

on how individuals identify with the President of the United States; individuals report better health when there is an alignment between their political preferences and the party of the President.

We then proceed to investigate the potential mechanisms behind our main findings. We acknowledge that there could be multiple factors simultaneously contributing to these results. Furthermore, our reduced form analysis does not allow us to disentangle the extent to which each mechanism we investigate can explain the documented results. Nonetheless, our examination still yields interesting insights into our main findings.

We start by providing evidence that divergent political views and sentiments play a potentially important role in our setting. To do so, we first focus our attention on sudden deaths that can be associated with stress and anxiety. If political sentiments are driving our findings, we should find a sudden increase in this type of mortality rate around crucial political events, when political polarization is greater and therefore also its short-term expected impact on health. Consistent with this hypothesis, we document a sharp increase in deaths from cardiovascular diseases and other external causes of death (such as suicides and drug overdoses) in losing counties in the months following turnover elections. Furthermore, we document an increase in mortality in politically polarized counties during periods of high political stress, that it is the period before a Presidential election [21].

While exposure to stress may indeed have an immediate, short-term impact on mortality rates [22], sustained exposure to this factor and political sentiments can also affect the mental health and lifestyle of the individuals, contributing to medium-term, cumulative negative effects on health outcomes [23]. This may potentially explain the long-lasting effects of our findings.

In line with this hypothesis, we find that individuals in losing counties report a higher probability of having mental issues related to stress, depression, and problems with emotions when the President of the opposite party is in charge. Furthermore, we argue that the frustration and depression associated with losing an election can impact individuals' behavior as well. Previous research indeed suggests that these negative sentiments adversely affect individuals' social interactions [24,25]. On the other hand, social isolation and limited involvement in community life are associated with worse health and mortality in both the short- and medium-term [23,26–29].

To investigate whether losing an election impacts individuals' social behavior, we proxy the degree of communities' involvement in social activities using the number of membership and non-profit organizations per 10,000 population [30]. Consistent with our hypothesis, we document in losing counties a decrease in these types of organizations after turnover Presidential elections. Overall, these results suggest that the encounter with these factors during and following transitional political phases may contribute to the development of chronic health conditions or worsen existing ones, ultimately leading to higher mortality rates over time.

At the end of the paper, we shed further light on the potential mechanisms behind our main finding and investigate whether changes in economic outcomes or in other economic policies benefiting the related partisan group of the population can also explain our results. However, consistent with [11], we do not find any evidence that the partisanship of the community affects, in our setting, county establishment formation, employment, wages, house prices, or transfers to households.

**Previous literature and contribution.** Our paper is related to the growing literature analyzing the economic cost and benefits of a change in political leadership on economic outcomes. [31] study turnover elections across countries and find that turnovers improve country performance. They also show that this effect is not driven by differences in the characteristics of the politicians, or by the fact that the level of government intervention in the economy

increases after the elections. Other papers argue that turnover elections could have a detrimental effect on economic outcomes due to politicians' loss of experience [32,33] and policy uncertainty [34].

Within this literature, recent papers exploit turnover elections to understand how political partisanship and representation shape economic outcomes. On the one side, political partisanship has been shown to affect expectations about future economic prosperity, therefore affecting consumption patterns [15,17,20,35], the decision to start a new business and investors' beliefs [16,36,37], and fertility [18]. On the other hand, [11] linked a change in political leadership to changes in sentiments regarding government economic policies; however, consistent with [38], they fail to find any evidence that these changes are linked to substantial changes in consumption patterns. Along the same line, our paper shows that changes in political leadership affect individuals' sentiments, significantly impacting their health but without any significant effects on economic outcomes.

Political partisanship in the United States has been recently associated with health outcomes during the COVID-19 pandemic and the Trump presidency. On this point, [39] provide evidence that areas with more Republicans exhibited less social distancing during the pandemic and identify significant differences between Republicans and Democrats in self-reported social distancing behavior and beliefs about COVID-19 risk and the severity of the pandemic. [40] use geo-tracking data of 15 million smartphones to show that Republican counties engaged in less physical distancing and it led to subsequent higher COVID-19 infection and fatality rates in these areas. Similar results have been presented by [41], highlighting the impact of the conservative Fox News Channel in the USA on physical distancing during the COVID-19 pandemic. [42] show that partisanship is the most important predictor of mask use during the COVID-19 pandemic and local policy interventions do not offset this relationship. [43] examine the impact of the January 6, 2021 Capitol riot on risk avoidance behavior and COVID-19 spread, finding increased stay-at-home behavior among DC residents, no evidence of increased local COVID-19 spread in DC, but some evidence of increased COVID-19 cases in counties that had higher numbers of residents travel to the Capitol protest. Unlike studies focused on specific health behaviors and exploiting COVID-19 as an exogenous event, our paper quantifies the overall mortality burden of political misalignment, making it more relevant for broad public health policy. Our calculation of economic costs using FEMA's value of statistical life provides a concrete policy-relevant metric absent from most previous research on partisan health differences.

Other papers in the political science literature analyze how Presidential elections affect the health of individuals. [21] show that the mental health of the individuals got worse during the 2020 Presidential campaign. However, this effect was smaller for individuals supporting the winning candidate, Joe Biden. [44] find opposite results with respect to ours; they show that the suicide rate when a state supports the losing candidate will tend to be lower than if the state had supported the winning candidate. They argue that being around others who also supported the losing candidate may indicate a greater degree of social integration at the local level, thereby lowering relative suicide rates. Their focus on suicides, distinct from the broader scope of overall mortality rates, underscores a divergence in the emphasis of our respective studies. Furthermore, their focus on the period from 1981 to 2005 differs from ours, as during that time-frame, political polarization was not as pronounced [1–4]. Therefore, it is possible that during their period of analysis, the benefits of social integration were higher than the emotional costs of losing an election. [45] find that elections strongly affect the happiness of partisan losers, but this effect dissipates within a week after the election. Our study demonstrates that health effects are not transient but persist over years, suggesting a different and

more concerning mechanism than temporary emotional responses. Our long-term GSS analysis spanning from 1973 provides historical context missing from most contemporary studies, offering evidence that the health-politics relationship has been consistent across multiple administrations.

Finally, [46] identified a positive correlation between mortality rates and partisan losses. In a similar vein, [47] and [48] use regression analysis to demonstrate a correlation between health metrics and voting patterns, with voting outcomes serving as the dependent variable. Our study differs from these works by providing causal, unidirectional estimates of the impact of a shift in political power on mortality, leveraging turnover elections as natural experiments. In addition to this, our unique county-race-year level analysis of black mortality following Obama's election provides a within-county comparison that controls for unobserved county characteristics, a methodological improvement over studies that rely solely on cross-county comparisons. More importantly, we investigate the specific potential mechanisms driving these effects, emphasizing the role of political sentiments and social isolation while ruling out changes in economic characteristics as a possible explanation. Unfortunately, we do not empirically assess their relative contribution to the observed effects on mortality, and we cannot exclude the possibility that other mechanisms also play a role in this setting. Overall, our analysis offers a more comprehensive understanding of the relationship between health outcomes and political leadership.

## 2 Methodology

### 2.1 Database

We collect information from different sources. This section describes the variables that we use in our empirical analysis and their sources.

**Death certificates.** Our main proxy for communities' health is the county age-adjusted mortality rate. This measure presents several advantages. First, it is not self-reported and is therefore not subject to the conventional problems of survey data (e.g., [49,50]). Second, because our data exhaustively cover the whole U.S. population, our analysis allows us to make general estimates of the overall health costs related to a change in political leadership. On the other hand, mortality is an extreme health outcome, and changes in a person's mental or physical well-being do not always involve the death of the individual. Therefore, our estimates will underestimate the true effect on health.

Our source of information for this variable is the U.S. Centers for Disease Control (CDC). It provides us with detailed administrative information on the universe of death certificates for U.S. residents. Each death certificate provides detailed information on the deceased's demographic characteristics, such as age, race, and place of residence.

We use this information to compute age-adjusted mortality rates for United States counties. To do so, we count the number of deaths by county, age categories, and year and merge it with population information from the National Cancer Institute's Surveillance, Epidemiology End Results (SEER) Program. The yearly age-adjusted mortality rates (per 100,000 population) are a weighted average of the crude death rates across age categories within a county, where the shares of the overall population in each age category are used as weights.

The age categories we consider in our analysis are less than one year, 1-4 years, 5-14 years, 15-24 years, 25-34 years,...and older than 85 years [51]. The weights are computed considering U.S. population shares in the year 2000. We report more information about the age-adjusted mortality variable in the Online Appendix and the weights we use in S1 Table.

**Political preferences.** We collect information from the MIT Election Data and Science Lab to identify the political preferences of the communities. The database provides county-level

information on the Presidential election voting from the year 2000. This approach has been used in recent empirical research (e.g., [11,18,19]).

Our main measure of political partisanship is the county-level vote share for the Republican and Democratic parties, respectively. In an alternative exercise, we also identify an electoral loss with a dummy variable equal to one if the county's political preference is for the Democratic (Republican) party and the party of the President is Republican (Democrat). We define the county political preference for this variable using as threshold 50% of vote share for the Democratic party.

We next consider in our analysis a measure of county political polarization, that is the absolute distance between the political preferences of each county and the neighboring counties (i.e. all counties sharing a border). The distance is measured as the difference between the historical vote share for the Democratic party of a county and the average vote share for the same party in the neighboring counties, as reported in the following Eq (1):

$$Polarization_c = \mid Share\ Vote_c - Average\ Vote\ Share\ Neighbour_c \mid \tag{1}$$

This variable ranges between 0 and 1 and a higher value is associated with a greater distance in the political preferences of a county with respect to their neighbors. More specifically, in our setting, this variable can hypothetically assume a maximum value of 1 if the whole county supports the Democrat party and all the other neighboring counties comprehensively support the Republican party, and vice-versa. It assumes a value equal to 0 if there is not any distance in the political preferences of the two counties (both support the Democrat party with the same intensity).

**Membership associations.** We consider the total number of membership associations per 10,000 population as a measure of individuals' social involvement in the community [30]. More specifically, we collect yearly information on membership associations from the National Establishment Time Series (NETS) database. This database provides us with information on the universe of establishments in the United States; therefore, it allows us to observe the dynamics of membership organizations in each county. These include civic organizations, bowling centers, golf clubs, fitness centers, sports organizations, religious organizations, political organizations, labor organizations, business organizations, and professional organizations. Membership associations are identified using the NAICS codes (813410, 713950, 713910, 713940, 711211, 813110, 813940, 813930, 813910, and 813920).

**BRFSS.** We collect survey information to measure individuals' health. This information comes from the Behavioral Risk Factor Surveillance System (BRFSS). BRFSS completes more than 400,000 interviews each year and it is for this reason the largest continuously conducted health survey system in the world.

The information available in this database allows us to follow previous literature (e.g., [52]) and measure health through a survey question measuring self-rated general health status. More specifically, we rely on the following question to quantify the health of an individual: *"Would you say that in general your health is excellent, very good, good, fair, poor?"*. This measure has been shown to outperform other objective health measures and to be highly correlated with mortality even after controlling for other baseline demographic characteristics (e.g., [53,54]). We assigned a maximum value of 4 to "Excellent" and a minimum value of 0 to "Poor".

We also rely on the following question to evaluate the mental health of the individuals: *"Now thinking about your mental health, which includes stress, depression, and problems with emotions, for how many days during the past 30 days was your mental health not good?"*. The variable spans from a minimum of 0 to a maximum value of 30.

While this database does not provide information on the political preferences of the individuals, it provides us with information on their county of residence, which we use to proxy their political preferences. Also, another concern with this database is that it does not allow us to follow individuals over time.

**GSS.** The General Social Survey (GSS) delivers cross-sectional annual (or bi-annual) information starting from the year 1973. It provides us with information on the health of the individuals, that it is possible to measure considering the self-rated general health status measure *"Would you say that in general your health is excellent, very good, good, fair, poor?"*. Importantly for our purpose, the database also provides us with information on the political preferences of the individuals. This information can be obtained by the following question *"Generally speaking, do you usually think of yourself as a Republican, Democrat, Independent, or what?"*.

A shortcoming of this database is that its public version does not provide information on individuals' county of residence and does not allow us to follow individuals over time. The restricted version of the GSS does report geographic identifiers, including county of residence. While we do not use this in our current analysis, it is worth noting that the restricted-use GSS data could potentially allow for controlling for county-fixed effects in our empirical analysis. Also, its size is relatively small with respect to the other databases that we consider in our analysis. The total number of observations is indeed equal to 68,846. Once we remove observations with missing values for our variables of interest (health, political preferences, age, income, marital status, and sex), we obtain a final database composed of 43,943 observations.

**County characteristics.** We collect information on county economic characteristics from the Bureau of Economic Analysis (BEA). More specifically, we collect from this source information on population, income per capita, and current transfers to individuals. We get information on unemployment from the Local Area Unemployment Statistics (LAUS) and on the number of establishments, employees, and total wages from the County Business Patterns (CBP) program of the Census Bureau. We also collect information on house prices from the Federal Housing Finance Agency (FHFA).

**Summary statistics.** In our empirical analysis, we use information at different levels of aggregation. We first merge information on mortality with information on county-level economic characteristics for the whole spanning period for which information is available. We next focus separately on the surrounding periods of the turnover Presidential elections of Barack Obama and Donald Trump. We use BRFSS survey information to test the impact of a change in political leadership on general and mental health. Finally, we use the GSS in a robustness check to provide evidence about the external validity of our results. We report the summary statistics of each sample in Table 1. A detailed description of each variable is available in S2 Table.

## 2.2 Mortality and political partisanship

**Empirical approach.** For identification purposes, we follow recent empirical literature (e.g., [11,18–20]). In a first approach, we consider the whole spanning period for which our data are available, the period 2000-2019. Furthermore, for this exercise, we only include in the sample counties that never change their political preferences during this spanning period. It restricts our sample to 2,416 counties. The total number of counties in the United States is 3,143. Our sample for this exercise represents approximately 77% of all US counties, providing broad geographic coverage and ensuring that our findings are nationally representative.

**Table 1.** Descriptive statistics.

| Variables | (1) Count | (2) Mean | (3) SD | (4) p25 | (5) p50 | (6) p75 |
|---|---|---|---|---|---|---|
| *Panel A: County level analysis* | | | | | | |
| Mortality | 49,016 | 833.5517 | 161.2961 | 731.0636 | 829.3458 | 932.1209 |
| Electoral Loss | 49,016 | 0.4232 | 0.4941 | 0.0000 | 0.0000 | 1.0000 |
| Population (log) | 48,316 | 10.1608 | 1.4419 | 9.2321 | 10.0755 | 10.9962 |
| Income (log) | 48,316 | 10.3895 | 0.3018 | 10.1772 | 10.3739 | 10.5802 |
| Unemployment (%) | 48,982 | 5.8913 | 2.6761 | 4.0000 | 5.3000 | 7.2000 |
| Transfers (log) | 48,316 | 8.8377 | 0.3596 | 8.5790 | 8.8740 | 9.1159 |
| Wages (log) | 48,992 | 12.0953 | 2.0422 | 10.8845 | 12.0665 | 13.2299 |
| Establishment (log) | 48,992 | 6.3444 | 1.4572 | 5.3613 | 6.2246 | 7.1577 |
| Employment (log) | 48,992 | 8.6931 | 1.8595 | 7.5673 | 8.6735 | 9.7550 |
| HPI (log) | 36,969 | 4.8747 | 0.1844 | 4.7322 | 4.8544 | 4.9832 |
| Polarization | 48,936 | 0.0599 | 0.0592 | 0.0194 | 0.0431 | 0.0809 |
| *Panel B: Obama sample* | | | | | | |
| Mortality | 28,017 | 824.8195 | 158.2258 | 721.3644 | 816.4392 | 919.5724 |
| Republicans | 28,017 | 0.5681 | 0.1381 | 0.4780 | 0.5721 | 0.6686 |
| Population (log) | 27,540 | 10.2676 | 1.4519 | 9.3319 | 10.1580 | 11.1022 |
| Income (log) | 27,540 | 10.3377 | 0.2278 | 10.1898 | 10.3194 | 10.4623 |
| Unemployment (%) | 28,008 | 4.8529 | 1.6838 | 3.7000 | 4.6000 | 5.6500 |
| Transfers (log) | 27,540 | 8.8047 | 0.2770 | 8.6312 | 8.8240 | 9.0057 |
| Establishments (log) | 28,006 | 6.4667 | 1.4838 | 5.4596 | 6.3306 | 7.3185 |
| Employment (log) | 28,006 | 8.8106 | 1.9201 | 7.6889 | 8.7869 | 9.9006 |
| Wages (log) | 28,006 | 12.1896 | 2.1026 | 10.9670 | 12.1519 | 13.3717 |
| HPI (log) | 21,609 | 4.8907 | 0.1587 | 4.7836 | 4.8641 | 4.9724 |
| Membership | 27,540 | 39.54323 | 14.4029 | 30.0621 | 37.3527 | 46.3171 |
| *Panel C: Trump sample* | | | | | | |
| Mortality | 24,904 | 805.5837 | 163.8440 | 701.3565 | 796.5535 | 901.9814 |
| Democrats | 24,904 | 0.3154 | 0.1522 | 0.2037 | 0.2832 | 0.3991 |
| Income (log) | 24,480 | 10.5870 | 0.2388 | 10.4274 | 10.5591 | 10.7160 |
| Population (log) | 24,480 | 10.2811 | 1.4811 | 9.3078 | 10.1560 | 11.1314 |
| Unemployment(%) | 24,896 | 5.5013 | 1.9476 | 4.2000 | 5.3000 | 6.5000 |
| Transfers (log) | 24,480 | 9.1164 | 0.2316 | 8.9809 | 9.1359 | 9.2717 |
| Employment (log) | 24,894 | 8.8530 | 1.8375 | 7.6866 | 8.7826 | 9.9270 |
| Wages (log) | 24,894 | 12.4360 | 1.9900 | 11.1899 | 12.3477 | 13.5877 |
| Establishment (log) | 24,894 | 6.4458 | 1.5015 | 5.4116 | 6.3044 | 7.3139 |
| HPI (log) | 19,199 | 4.9484 | 0.1810 | 4.8321 | 4.9327 | 5.0515 |
| Membership | 24,480 | 37.6049 | 14.1509 | 28.6818 | 35.1090 | 43.6189 |
| *Panel D: BRFSS* | | | | | | |
| General Health | 2,929,186 | 2.4865 | 1.0968 | 2.0000 | 3.0000 | 3.0000 |
| Mental Health | 2,929,186 | 3.4846 | 7.7411 | 0.0000 | 0.0000 | 2.0000 |
| Age | 2,929,186 | 53.1837 | 16.5709 | 41.0000 | 53.0000 | 65.0000 |
| Married | 2,929,186 | 0.5565 | 0.4968 | 0.0000 | 1.0000 | 1.0000 |
| Female | 2,929,186 | 0.6014 | 0.4896 | 0.0000 | 1.0000 | 1.0000 |
| *Panel E: GSS* | | | | | | |
| General Health | 43,943 | 1.9983 | 0.8355 | 2.0000 | 2.0000 | 3.0000 |
| Democrat | 43,943 | 0.1645 | 0.3707 | 0.0000 | 0.0000 | 0.0000 |
| Democratic Party | 43,943 | 0.4671 | 0.4989 | 0.0000 | 0.0000 | 1.0000 |
| Age | 43,943 | 45.8911 | 17.2505 | 32.0000 | 43.0000 | 59.0000 |
| Female | 43,943 | 0.5503 | 0.4975 | 0.0000 | 1.0000 | 1.0000 |
| Married | 43,943 | 0.5253 | 0.4994 | 0.0000 | 1.0000 | 1.0000 |

**Notes**: This table shows descriptive statistics of our analysis. See S2 Table for a detailed description of every variable and its source.

In this way, our identification strategy exploits counties entering and exiting political align-
ment with the party of the President. The period of analysis includes the shift from the pres-
idency of the Republican George W. Bush to the Democrat President Barack Obama in 2008
and from the Democrat President Obama to the Republican President Donald Trump in 2016.

To test whether a shift in political leadership affects the health outcomes of the communi-
ties in the losing counties, we estimate the following Equation:

$$Mortality_{c,t} = \beta Electoral\ Loss_{c,t} + \Gamma X_{i,t} + \eta_c + \theta_{t,s} + \epsilon_{i,t} \qquad (2)$$

In Eq (2), *Mortality* is the age-adjusted mortality rate in county *c* at year *t*. *Electoral Loss*$_{c,t}$
is a dummy variable equal to one if the county's political preference is for the Democratic
(Republican) party and the party of the President is Republican (Democrat). We define the
county partisanship using as threshold 50% of the vote share for the Democratic party. *X* is
a matrix of control variables ((log) population, (log) income per capita, and unemployment
rate). $\eta_c$ and $\theta_{t,s}$ are respectively county and state-year fixed effects.

We report estimation results in Table 2. The first column shows estimation results when
we consider our simplest model that only includes county and year-fixed effects. Column (2)
shows estimation results when we control for state times year fixed effects. In the last column,
we also include the time-varying controls for the county characteristics.

The coefficient of interest is always positive and statistically significant. According to the
results reported in the last column, we find in losing counties a yearly increase in the age-
adjusted mortality rate of respectively 7 units (7 deaths per 100,000 population). Consid-
ering that the estimated value of a life in the United States according to the Federal Emer-
gency Management Agency (FEMA) is US $7.5 million in 2020 and that the average county
population is equal to 104,487, the effect is economically meaningful.

**A comparison with previous studies.** Our results are comparable, and in some cases sub-
stantially higher, to the results of other papers analyzing mortality dynamics in the United
States. For example, [55] shows that a one percentage point increase in the state unemploy-
ment rate decreases the death rates by 4.6 individuals per 100,000 population. Similarly, [56]
leverage spatial variation in the severity of the Great Recession across commuting zones and
estimate that an increase in the unemployment rate of the magnitude of the Great Reces-
sion reduces the age-adjusted mortality rate by 18 individuals per 100,000 population, with
effects persisting for at least 10 years. [57] find that one standard deviation change in banking

**Table 2. Baseline.**

| Variables | (1) Mortality | (2) Mortality | (3) Mortality |
|---|---|---|---|
| Electoral Loss | 10.1545*** | 7.5226*** | 7.0844** |
|  | (2.8978) | (2.3138) | (2.8108) |
| County FE | Yes | Yes | Yes |
| Year FE | Yes | Yes | Yes |
| State-Year FE | No | Yes | Yes |
| County controls | No | No | Yes |
| Observations | 49,016 | 49,016 | 48,302 |
| Adjusted R-squared | 0.670 | 0.676 | 0.679 |

**Notes**: This table shows regression results for Eq (2). *Mortality* is the dependent variable and is the age-adjusted
mortality rate of a county. Standard errors are double clustered at the county and year level. ***, **, and * denote
significance at 1, 5, and 10 percent level respectively. See S1 Sect of the Online Appendix for a detailed description of
every variable.

deregulation activities decreases overall mortality rates by 5 individuals per 100,000 population. [58] analyze the effect of temperature shocks on mortality and find that an 85°F – 90°F day increases the mortality rate in the coldest counties by 1.8 deaths per 100,000 population. [59] find that Medicaid expansions led to a significant decline in mortality equivalent to 19.1 deaths per 100,000 population. On the other side, [60] investigate the effects of Medicare's introduction in 1965 on elderly mortality over its first decade, revealing no significant impact on mortality but a notable reduction in out-of-pocket medical expenditure risk among the elderly population. Similarly, using a randomized controlled design, [61] document no effect of access to public health insurance for a group of uninsured low-income adults on mortality. However, the authors also note that this result is not surprising, considering the extremely low mortality rate in their population.

**The two presidential turnover elections.** Our identification strategy is based on the two Presidential turnover elections of President Barack Obama in 2008 and President Donald Trump in 2016. An important assumption behind the validity of our difference-in-differences approach is that Republican and Democratic counties have parallel (counterfactual) trends in post-treatment periods. In this paragraph, we analyze this assumption graphically and empirically.

To graphically analyze the mortality trends of Republican and Democratic counties, we split the sample based on the median vote share for the Democratic party during the two turnover elections. We then demean the trends with respect to the average mortality value before the elections and inspect the time-series graphs. Overall, Fig 1 shows that, before the turnover elections, the two groups have similar mortality patterns. We also document divergent patterns after the elections.

Mortality patterns also follow similar patterns in the year of the elections, even if other papers show that individuals' health deteriorates during the elections (e.g., [21]). However, these pre-election effects are unlikely to invalidate our difference-in-differences approach since campaign stress is likely to be similar across partisan counties before the election, with divergence only occurring after the election outcome is known. In Fig 1(a), there is a noticeable dip in mortality trends for both Democratic and Republican counties between 2006 and 2007, before the election of Obama. However, it is also possible to notice a more substantial decrease in the mortality rate in Democratic counties. We test in a robustness check whether this difference is statistically significant and whether it leads to a violation of the parallel trend assumption.

To provide further evidence on the validity of our approach, we test separately the impact of the 2008 and 2016 turnover elections on mortality patterns by estimating for the two alternative periods the following Eq (3):

$$Mortality_{c,t} = \sum_{t=-3, t \neq ElectionYear}^{t=+5} \beta_t Political\ Preferences_c \times Year_t + \eta_c + \theta_{t,s} + \epsilon_{i,t} \qquad (3)$$

In Eq (3), *Political Preferences* is the share of votes for the opposition political party (Republican and Democratic, alternatively) of county $c$ during the 2008 and 2016 Presidential elections. $\eta_c$ and $\theta_{t,s}$ are respectively county and state-year fixed effects. In this model, we interact *Political Preferences* with a complete set of year dummies using the election years as the reference years. We consider three lags and five leads around the election years. We exclude the fifth year after the Trump election because the mortality data were not available when we start to write the paper and also to avoid our results could be affected by the Covid-19 epidemiological crisis. Thereby, the coefficients $\beta_t$ report the differential effect of *Political*

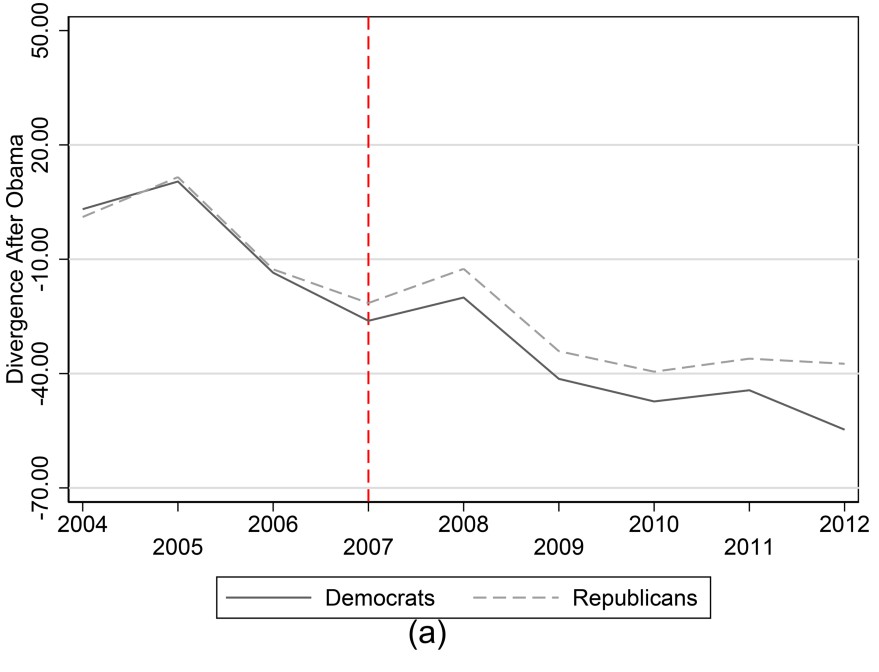

(a)

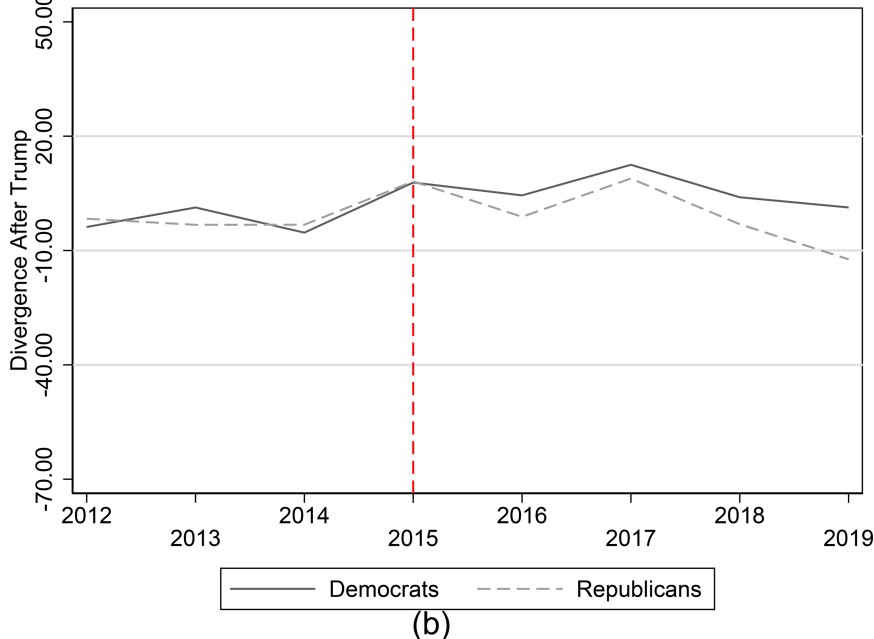

(b)

**Fig 1. Parallel trend assumption.**

*Preferences* on age-adjusted mortality rates for a particular year compared to the years before the Presidential elections.

We present the $\beta$ coefficients estimated from Eq (3) and the 95% confidence intervals in Fig 2. The graphs provide several important results. First, the yearly point estimates show significant and positive effects for every year after the two elections for one unit increase in the

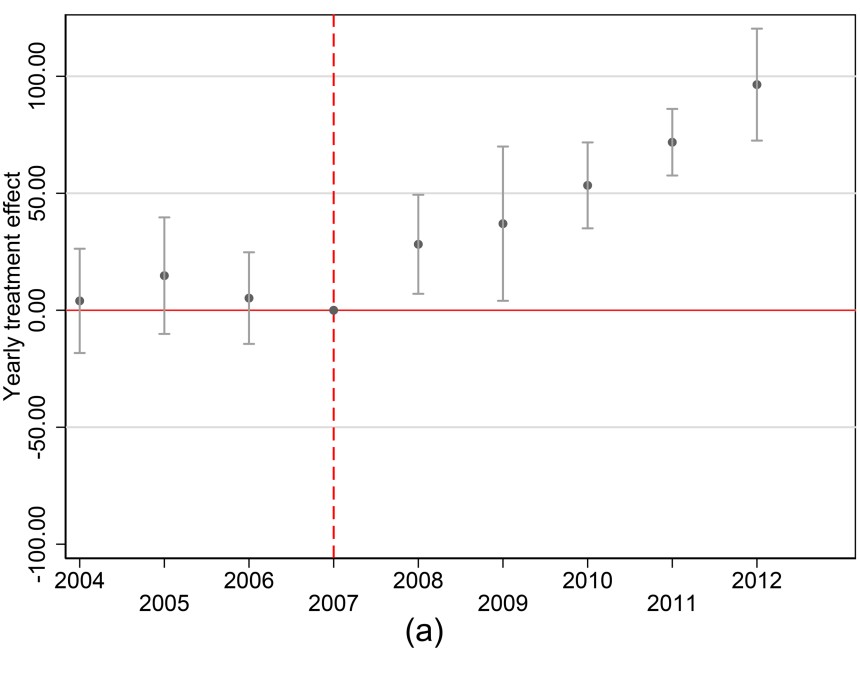

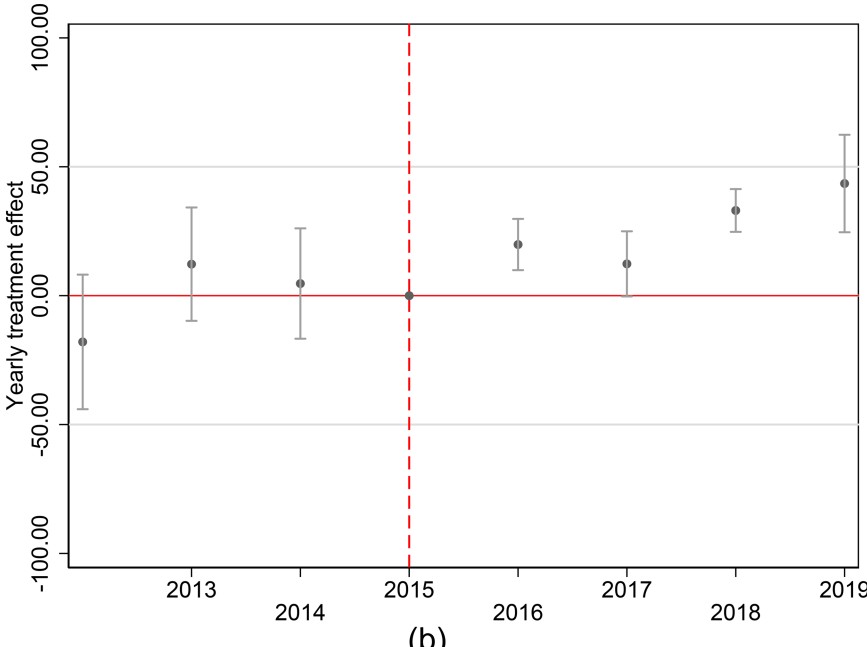

**Fig 2. Yearly treatment coefficients.**

Republican (Democrat) vote share (from 0 % to 100 %). In terms of economic magnitude, we estimate a coefficient of about 50 (50 deaths per 100'000 population) for the 2007 Presidential election of Barack Obama and 25 (25 deaths per 100,000 population) for the 2016 Presidential election of Donald Trump.

Also, we find that the yearly treatment effects are not significant before the two elections. In combination with Fig 1, we further consider the absence of significant effects in the pre-election period to indicate that the U.S. counties are likely to follow parallel trends in terms of mortality across the vote shares distribution [62]. Recent studies have warned about the limitations of pre-event trend testing and raised concerns about the low power to detect meaningful violations of parallel trends using this approach [63–65]. To test the robustness of our test, we conduct a sensitivity test following the approach proposed by [66]. Importantly, S1 Fig shows that the fixed length confidence intervals are similar to those from our baseline estimators when allowing for violations of parallel trends that are approximately linear and for larger degrees of possible non-linearity in the violation of parallel trends.

**Difference-in-differences results.** Event study estimates from Eq (3) identify treatment effects over time and provide evidence that the parallel trend assumption behind the validity of our approach is likely to hold. Since we show the effect is long-lasting and persistent, we consider in this paragraph a more efficient difference-in-differences approach and estimate separately the following Eq (4) for the two turnover Presidential elections:

$$Mortality_{c,t} = \beta Political\ Preferences_c \times Post_t + \Gamma X_c \times Post_t + \eta_c + \theta_{t,s} + \epsilon_{i,t} \qquad (4)$$

In this setting, the variable *Post* is a dummy variable equal to 1 after the 2008 or 2016 Presidential elections and 0 otherwise. We include in our model $X_c$, a set of county characteristics. We measure these variables as in 2007 and 2015 respectively, since including time-varying control variables that could also be affected by the treatment could interfere with our estimates [67]. More specifically, we control for county population (log), unemployment rate, and income per capita (log). The $\beta$ coefficient measures the effect on the age-adjusted mortality for one unit change in the opposition partisan vote share (from 0 % to 100 %) after the 2008 and 2016 Presidential elections relative to the run-up periods 2004-2007 and 2012-2015.

We report estimation results from Eq (4) for the Obama Presidential election in Panel A of Table 3, using as independent variable of interest the Republican vote share. Column (1) shows the results of our simplest specification, where we only control for county and year-fixed effects. In line with our previous results, we find a positive effect of the Republican vote share on the age-adjusted mortality rates after the election of Obama. More specifically, a unit increase in the vote share for the Republican party (from 0 to 100%) increases mortality by 44 units (44 deaths per 100,000 population). It implies that one standard deviation increase in the Republican vote share (a change of 14 %) increases age-adjusted mortality by 6 units (6 deaths per 100,000 population), which represents an increase of 0.7 % with respect to the average age-adjusted mortality rate. Column (2) shows estimation results when we include in our model state times year fixed effects. In this specification, we exploit with-in-state variation in partisan preferences and we get similar results. Finally, in the last column, we propose our preferred specification and control for the set of county characteristics. Again, our coefficient of interest is still positive and statistically significant. In terms of magnitude, it is slightly higher with respect to the results reported in the first column and equal to 44.6.

We report in Panel B estimation results from Eq (4) using as an independent variable of interest the Democrat vote share and as the event the Presidential election of Donald Trump in 2016. We also find a positive and statistically significant effect on the age-adjusted mortality rates. According to the results reported in the last column, a one-unit increase in the vote share for the Democrat party (from 0% to 100%) increases mortality by 30 units (30 deaths per 100,000 population). In terms of magnitude, one standard deviation increase in the Democrat vote share (a change of 15 %) increases mortality rates by 0.9 % with respect to the average.

**Table 3**. Difference-in-differences.

| Variables | (1) Mortality | (2) Mortality | (3) Mortality |
|---|---|---|---|
| *Panel A: Obama and Mortality* | | | |
| Post × Republicans | 44.2371*** | 51.3716*** | 44.6786** |
| | (11.6102) | (14.3100) | (15.2342) |
| County FE | Yes | Yes | Yes |
| Year FE | Yes | Yes | Yes |
| State-Year FE | No | Yes | Yes |
| Interacted controls | No | No | Yes |
| Observations | 28,017 | 28,017 | 27,540 |
| Adjusted R-squared | 0.701 | 0.706 | 0.703 |
| *Panel B: Trump and Mortality* | | | |
| Post × Democrats | 23.2593* | 27.4390** | 30.9178** |
| | (10.5671) | (11.4452) | (12.3927) |
| County FE | Yes | Yes | Yes |
| Year FE | Yes | Yes | Yes |
| State-Year FE | No | Yes | Yes |
| Interacted controls | No | No | Yes |
| Observations | 24,904 | 24,904 | 24,480 |
| Adjusted R-squared | 0.747 | 0.747 | 0.747 |

**Notes**: This table shows regression results for Eq (4). *Mortality* is the dependent variable and is the age-adjusted mortality rate in the county. Standard errors are double clustered at the county and year level. ***, **, and * denote significance at 1, 5, and 10 percent level respectively. See S1 Sect of the online appendix for a detailed description of every variable.

## 2.3 Mechanisms

The previous section provides evidence that a change in political leadership affects health outcomes. We recognize that multiple factors could simultaneously contribute to this result. Furthermore, our reduced form analysis does not provide us with the capability to disentangle the extent to which each mechanism we investigate can explain the documented results. However, in this section, we still highlight interesting potential underlying mechanisms behind our main finding.

First, we argue that divergent political views with respect to the party of the President and the lack of political representation can increase stress and anxiety and decrease individuals' social interactions in the community with a potential negative effect on physical and mental health. These factors can influence mortality rates not only in the short-run [22], but also in the medium- and long-run [23].

To test this hypothesis, we look at the impact of a change in political leadership on the number of membership and non-profit organizations, mental health, and specific types of sudden deaths that can be caused by stress and anxiety around crucial political events. Also, in an alternative setting, we consider the impact of political polarization on health outcomes.

Next, we consider in our analysis that partisan biased beliefs about future economic growth and economic policies benefiting the related partisan group of the population can potentially affect households' welfare, consumption, and investment patterns (e.g., [15–17]), leading effectively to greater economic prosperity that can have a positive effect on community well-being. We investigate this point by looking at changes in economic outcomes and household transfers in losing communities after turnover elections.

### 2.3.1 Political sentiments and social isolation.

**Sudden deaths.** To investigate whether political sentiments are at the core of our findings, we focus our attention on sudden deaths that can be caused by a high level of stress and

anxiety. In line with our hypothesis, we should document a sudden increase in this type of mortality rate around crucial political events when political animosity is greater and, therefore, also its impact on health.

We identify a sudden death that can be caused by stress and anxiety as a death caused by cardiovascular disease or by an external cause, as defined by the CDC. Deaths by cardiovascular disease are the main cause of death in the United States and include heart attacks, stroke, coronary heart diseases, diseases of the arteries, and high blood pressure. Deaths by external causes include suicides, homicides, poisoning (including drug overdose), and other vehicle accidents. These types of death have the characteristic that immediately increases in moments of high stress and anxiety.

Several papers in different fields support this claim. In the medical literature, acute psychological stressors trigger well-documented physiological responses, including increased cardiovascular reactivity, inflammatory responses, and dysregulation of the hypothalamic-pituitary-adrenal axis [23,68]. These mechanisms provide biological pathways connecting sudden political disappointment to acute health events. In this sense, our hypothesis is also linked to the "broken heart syndrome" (takotsubo cardiomyopathy), where sudden emotional stress can trigger cardiovascular events even in otherwise healthy individuals [69]. Furthermore, other papers in the political science literature argue that crucial political events, such as the Presidential elections themselves, can be defined as sociopolitical stressful events by previous research [22,70].

To test our hypothesis, we measure quarterly mortality rates for these types of death and estimate Eq (3) using a spanning period of three quarters before and five quarters after the Presidential turnover elections. The reference quarter is the one before the election of the president. Using a tighter spanning period to measure the mortality rates comes at the cost of a higher measurement error; in order to mitigate this problem, we consider only counties with at least 10 deaths during the analyzed spanning period. The reason we did not apply such a restriction to the other specifications is that we do not encounter such problems, as mortality rates are computed yearly, considering all types of deaths. Therefore, fewer than 1% of the county-year observations have fewer than 10 deaths. Additionally, we chose the number 10 because it is used as a threshold value by CDC WONDER, where death counts and rates are suppressed if the number of deaths is less than 10.

We graphically report the estimated coefficients in Fig 3. We document, just after the turnover elections, an increase in mortality in the losing counties. More specifically, one standard deviation increase in the vote share for the Republican (Democrat) party increases quarterly mortality rates for external causes of death or cardiovascular diseases by almost 1 unit (1 death for 100,000 population). This result is in line with the idea that exposure to stress can have an immediate effect on health outcomes [22].

**Mental health.** To further test the hypothesis that political attitudes and sentiments are at the core of our findings, we test whether turnover elections affect the mental health of individuals. Therefore, we collect information on individuals' mental health from a large survey, the BRFSS. Importantly, this database also provides us with information on the county of the interviewers; however, this information is only available in the database until the year 2012, and we can, therefore, only focus on the 2008 turnover election.

To test our hypothesis, we estimate the following Equation:

$$Mental\ Health_{j,t} = \beta Political\ Preferences_c \times Post_t + \Gamma X_{j,t} + \eta_c + \theta_t + \epsilon_{i,t} \qquad (5)$$

In Eq 5, the outcome variable of interest is the Mental Health of individual $j$ at time $t$ measured with a dummy variable equal to one if the number of days in the last month the

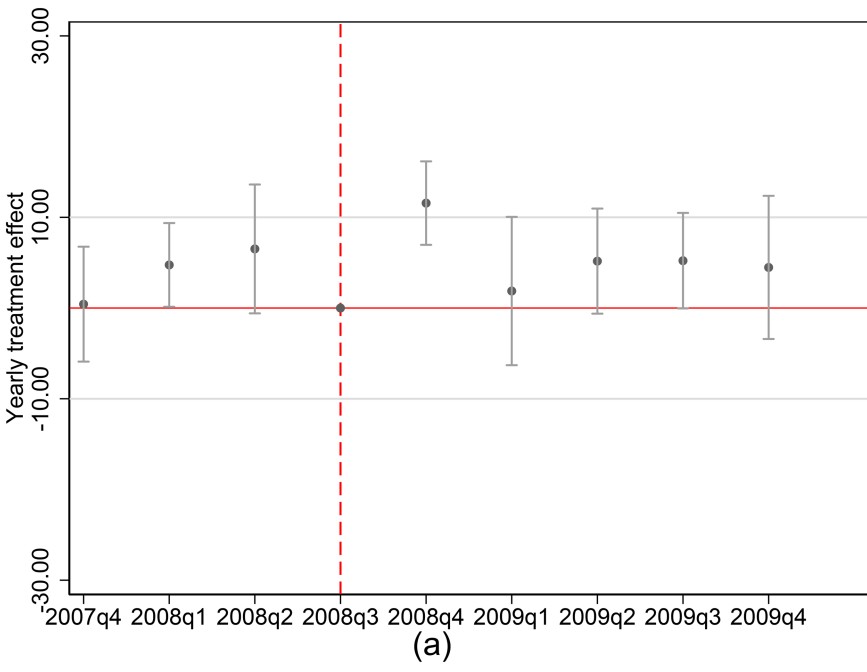

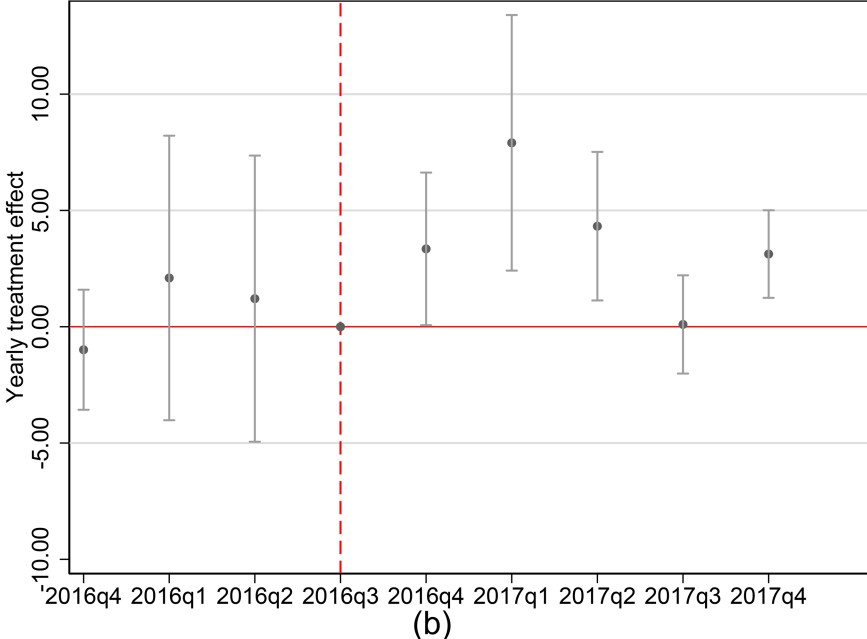

**Fig 3. Political sentiments.**

individual reported his/her mental health was not good is at least one. Mental health issues include stress, depression, and problems with emotions. We also include a set of individual control variables; more specifically, we control for age, age squared, a dummy variable equal to one if the individual is married, a dummy variable equal to one if the individual is female

and a set of dummy variables for each income category. Finally, we control for county and time-fixed effects. Again, we focus on the spanning period 2004-2012.

We report estimation results in Table 4. According to the second column, we find that one standard deviation increase in the share of Republicans in the county where the individual is located increases the probability that an individual reports at least a day with poor mental health by 0.25 % after the Obama election in Republican counties. In the last column, we show that our results do not change when we consider a continuous variable as an outcome. More specifically, we consider the natural logarithm of the reported number of days with poor mental health to deal with the skewness of the variable. Since this survey is not representative at the county level but rather at the state level, we show in S3 Table that our results hold when we measure our treatment variable (the share of Republican voters) at the state level. Furthermore, we use county-fixed effects to control for time-invariant county characteristics that might affect sampling.

**Political polarization.** If divergent political views affect health, we should document an increase in mortality in more politically isolated communities during periods of high political stress. In order to test this conjecture, we exploit the timing of the Presidential elections and

**Table 4**. Mental Health.

| Variables | (1)<br>Mental Health > 0 | (2)<br>Mental Health > 0 | (3)<br>Log(Mental Health) |
|---|---|---|---|
| Post × Republicans | 0.0144** | 0.0177** | 0.0325** |
| | (0.0061) | (0.0070) | (0.0148) |
| Age | | 0.0044*** | 0.0192*** |
| | | (0.0002) | (0.0005) |
| Age Squared | | −0.0001*** | −0.0003*** |
| | | (0.0000) | (0.0000) |
| Married | | −0.0395*** | −0.0950*** |
| | | (0.0010) | (0.0025) |
| Female | | 0.0957*** | 0.1798*** |
| | | (0.0010) | (0.0022) |
| Income = 2 | | −0.0440*** | −0.1735*** |
| | | (0.0019) | (0.0057) |
| Income = 3 | | −0.0834*** | −0.3094*** |
| | | (0.0020) | (0.0067) |
| Income = 4 | | −0.1107*** | −0.4096*** |
| | | (0.0023) | (0.0075) |
| Income = 5 | | −0.1424*** | −0.5207*** |
| | | (0.0023) | (0.0072) |
| Income = 6 | | −0.1615*** | −0.5954*** |
| | | (0.0026) | (0.0078) |
| Income = 7 | | −0.1811*** | −0.6687*** |
| | | (0.0028) | (0.0086) |
| Income = 8 | | −0.2179*** | −0.7703*** |
| | | (0.0031) | (0.0092) |
| County FE | Yes | Yes | Yes |
| Time FE | Yes | Yes | Yes |
| Observations | 2,929,186 | 2,929,186 | 2,929,186 |
| Adjusted R-squared | 0.00456 | 0.0700 | 0.0786 |

**Notes**: This table shows regression results for Eq (5). *Mental Health* is the dependent variable and is a dummy variable equal to 1 if (or the natural logarithm of) the number of days in the last month the individual reported his/her mental health was not good is at least one. Mental health issues include stress, depression, and problems with emotions. Standard errors are double clustered at the county and time level. ***, **, and * denote significance at 1, 5, and 10 percent level respectively. See S1 Sect of the online appendix for a detailed description of every variable.

estimate the following Eq (6):

$$Mortality_{c,t} = \beta Polarization_c \times Election\ Year_t + \Gamma X_{i,t} + \eta_c + \theta_{t,s} + \epsilon_{i,t} \qquad (6)$$

As we explained in the previous section, *Polarization* measures the distance between the political preferences of each county and their neighboring counties as the absolute difference between the vote share of the county for the Democratic party and the average vote share of the neighboring counties for the same party. *Election Year* is a dummy variable equal to 1 during the years of a Presidential election, and 0 otherwise In this setting, $\beta$ can be interpreted as the effect of political polarization on mortality during periods of high political stress.

We report the results in Table 5. In line with our hypothesis, we find an increase in mortality in politically isolated communities during the years of the Presidential elections. More specifically, according to the coefficient reported in the last column, we find that one standard deviation increase in the *Polarization* variable (an increase in 6 % in the distance) increases the age-adjusted mortality rates by 3.2 during the Presidential elections.

**Social isolation.** The frustration and depression associated with losing an election can also impact individuals' behavior. Previous research suggests indeed that these sentiments affect individuals' social interactions in the community [24,25]. On the other side, social isolation and limited involvement in community life have also been shown to be associated with worse health and mortality [26–29]. In this paragraph, we analyze whether losing an election is associated with a decline of social interactions in the losing communities. To do so, we estimate Eq (4) using as the outcome variable the number of membership associations divided by 10,000 population.

We report the results in Table 6. We find that the number of membership associations decreases in losing counties after a turnover election. More specifically, according to the results reported in the last column, we find that one standard deviation increase in the share of Republican (Democrats) decreases the outcome variable by 2.1% with respect to the average outcome variable after the election of Obama (Trump).

To offer additional evidence on the impact of turnover political elections on social capital, we gathered information on the number of non-profit organizations adjusted by total population from the National Center for Charitable Statistics (NCCS) [71,72], and provide consistent results in S4 Table. However, it's noteworthy that one of the two coefficients is not statistically significant at the conventional levels. A plausible explanation is that this variable might not be

**Table 5**. Political polarization.

| Variables | (1) Mortality | (2) Mortality | (3) Mortality |
|---|---|---|---|
| Election Year × Polarization | 55.6760 | 53.1879* | 54.7588* |
| | (42.6455) | (30.4575) | (31.2796) |
| County FE | Yes | Yes | Yes |
| Year FE | Yes | Yes | Yes |
| State-Year FE | No | Yes | Yes |
| County controls | No | No | Yes |
| Observations | 48,936 | 48,936 | 48,242 |
| Adjusted R-squared | 0.669 | 0.675 | 0.678 |

**Notes**: This table shows regression results for Eq (6). *Mortality* is the dependent variable and is the age-adjusted mortality rate in the county. Standard errors are double clustered at the county and year level. ***, **, and * denote significance at 1, 5, and 10 percent level respectively. See S1 Sect of the online appendix for a detailed description of every variable.

**Table 6**. Social isolation.

| Variables | (1) Membership | (2) Membership | (3) Membership |
|---|---|---|---|
| *Panel A: Obama and Membership Organizations* | | | |
| Post × Republicans | -5.0240** | -6.7955** | -5.8828*** |
| | (2.0898) | (2.1990) | (1.4547) |
| County FE | Yes | Yes | Yes |
| Year FE | Yes | Yes | Yes |
| State-Year FE | No | Yes | Yes |
| Interacted controls | No | No | Yes |
| Observations | 27,540 | 27,540 | 27,540 |
| Adjusted R-squared | 0.923 | 0.937 | 0.938 |
| *Panel B: Trump and Membership Organizations* | | | |
| Post × Democrats | 2.2736** | 1.6868 | -4.1403** |
| | (0.8262) | (0.9020) | (1.3848) |
| County FE | Yes | Yes | Yes |
| Year FE | Yes | Yes | Yes |
| State-Year FE | No | Yes | Yes |
| Interacted controls | No | No | Yes |
| Observations | 24,480 | 24,480 | 24,480 |
| Adjusted R-squared | 0.938 | 0.947 | 0.952 |

**Notes**: This table shows regression results for Eq (4) using as an outcome variable the number of membership organizations divided by 10'000 population (*Membership*). Standard errors are double clustered at the county and year level. ***, **, and * denote significance at 1, 5, and 10 percent level respectively. See S1 Sect of the online appendix for a detailed description of every variable.

a perfect measure, as not all organizations providing social capital are formalized non-profits that report to the IRS. Nevertheless, [73] find that it still serves as a reliable proxy for local civic opportunity.

These results sharply contrast with those of [44]. Specifically, they document a decrease in suicides following a Presidential election loss and argue that being surrounded by others who also supported the losing candidate may indicate a higher level of social integration at the local level, thus reducing relative suicide rates. However, in opposition to us, they did not formally test this hypothesis. A potential explanation for these differing results could be attributed to the different periods analyzed. Indeed, their focus on the period from 1981 to 2005 differs from ours, as during that timeframe, political polarization was not as pronounced [1–4]. Therefore, it is plausible that during their period of analysis, the benefits of social integration outweighed the emotional costs of losing an election.

**2.3.2 Economic prosperity.** Individuals have a more positive assessment of current and future economic conditions when the White House is occupied by the party they support [11,74]. These positive economic sentiments could lead to greater economic prosperity, even if previous research provides mixed evidence on the relationship between sentiments and economic activities [11,17].

A shift in political leadership can also affect economic policies, benefiting the related partisan group of the population. For instance, health care reform has been at the top of the nation's domestic policy agenda during the 2008 Presidential campaign, with the Republican and Democrat candidates proposing alternative plans to reform the health insurance system in the United States. While changes in economic policies are a possible explanation of our findings in the long-run, our results suggest a short-run reaction in the health of the communities, which is therefore unlikely to be explained by changes in economic policies. Related to

**Table 7. Economic prosperity.**

| Variables | Employment | Establishments | HPI | Wages | Trasnfers |
|---|---|---|---|---|---|
| *Panel A: Obama and Economic Performance* | | | | | |
| Post × Republicans | 0.1313 | 0.0494* | –0.0089 | 0.1910 | –0.0221 |
| | (0.1172) | (0.0254) | (0.0173) | (0.1521) | (0.0125) |
| County FE | Yes | Yes | Yes | Yes | Yes |
| Year FE | Yes | Yes | Yes | Yes | Yes |
| State-Year FE | Yes | Yes | Yes | Yes | Yes |
| Post × Controls | Yes | Yes | Yes | Yes | Yes |
| Observations | 27,528 | 27,528 | 21,150 | 27,528 | 27,540 |
| Adjusted R-squared | 0.941 | 0.999 | 0.911 | 0.919 | 0.985 |
| *Panel B: Trump and Economic Performance* | | | | | |
| Post × Democrats | –0.0091 | 0.2456* | 0.0031 | 0.3701 | –0.0088 |
| | (0.0113) | (0.1280) | (0.0136) | (0.1955) | (0.0070) |
| County FE | Yes | Yes | Yes | Yes | Yes |
| Year FE | Yes | Yes | Yes | Yes | Yes |
| State-Year FE | Yes | Yes | Yes | Yes | Yes |
| Post × Controls | Yes | Yes | Yes | Yes | Yes |
| Observations | 24,472 | 24,472 | 18,792 | 24,472 | 24,480 |
| Adjusted R-squared | 0.999 | 0.959 | 0.955 | 0.934 | 0.988 |

**Notes**: This table shows regression results for Eq (7), using as outcome the variables reported in the first row. Standard errors are double clustered at the county and year level. ***, **, and * denote significance at 1, 5, and 10 percent level respectively. See S1 Sect of the online appendix for a detailed description of every variable.

our previous example, the Affordable Care Act (ACA) reform law was only enacted in March 2010, allowing people to obtain health insurance only in subsequent years.

In this section, we analyze more in-depth whether a shift in political leadership affects economic prosperity in our setting as a possible additional channel linking political leadership and health. Therefore, we estimate the following Eq (7) for the two turnovers Presidential elections:

$$Outcome_{c,t} = \beta Political\ Preferences_c \times Post_t + \Gamma X_c \times Post_t + \eta_c + \theta_{t,s} + \epsilon_{i,t} \qquad (7)$$

In this setting, *Outcome* are alternative variables measuring local economic activity: the number of employees, the number of establishments, total wages, house price changes measured by the House Price Index (HPI), and transfers to individuals.

We report estimation results in Table 7. Consistent with [11], we do not find any evidence that economic sentiments affect economic prosperity or government transfers; if any, we find a slight increase in the number of establishments in losing counties, which works in the opposite direction of the hypothesis that a deterioration of the health is driven by worst economic outcomes.

## 2.4 Robustness checks

This section provides evidence that our main results are robust to a set of robustness checks. For space constraints, we report these results in the Online Appendix (OA) together with a detailed description of each exercise.

**Black individuals and Obama.** Following previous literature, we proxy individuals' political preferences using the county vote share. Ideally, we would like to know the political preferences of each deceased individual. However, the identity of an individual and his/her political

preferences are not available in the database. In this paragraph, we exploit a unique characteristic of the Obama Presidential election; most of the black voters (95 % of the black population) cast their ballot for the Democrat Barack Obama (Pew Research Center, 2009).

Exploiting this peculiarity, we build a new database at county-race-year level and compare black mortality rates with the mortality rates of other races (White, Hispanic, and Asiatic). This approach is particularly useful since it allows us to compare individuals living in the *same county* at the same period, and it helps to provide further evidence against the validity of our identification strategy. We, therefore, estimate the following Eq (8):

$$Mortality_{c,r,t} = \beta Black_r \times Post_t + \Gamma X_c \times Post_t + \eta_c + \theta_{t,s} + \iota_{r,s} + \epsilon_{i,t} \tag{8}$$

In this Equation, *Mortality* is the age-adjusted mortality rate in county $c$ of race $r$ at time $t$. *Black* is a dummy variable equal to 1 for the black race and 0 otherwise. Since there is great heterogeneity in mortality patterns by race across U.S. states [57], we control for state times race ($\iota_{r,s}$) fixed effects.

We report estimation results in S5 Table. According to the idea that a shift in political leadership affects health outcomes, we find a decrease in the mortality of black individuals. More specifically, according to the last column, black mortality rates decrease after Obama's election by 35 units.

We also show graphically the patterns of black mortality rates in S2 Fig; on the left, we show that after the election of President Barack Obama, there has been a decrease in the average mortality of black individuals with respect to the mortality of the other races, when there is not a clear pattern before the election of the new President. On the right, we report the yearly treatment effects from an estimated dynamic model similar to the one reported in Eq (3). The graph documents a decrease in the black age-adjusted mortality rates, thereby confirming our previous findings.

**Outliers.** We show that our results are not driven by a sub-set of observations. To do so, we report in S3 Fig the added variable plots from Eq (4) for the interaction term between *Political Preferences* and *Post* for the two turnover Presidential elections. It shows the influence of political preferences after the Presidential election on mortality while simultaneously accounting for the influence of all the other independent variables and fixed effects. Importantly, the figures do not indicate that outliers drive our estimated effects.

In order to provide further evidence that outliers do not affect our results, we estimate Eq (3) after winsorizing all the variables in the model at the first and the last percentiles. We report the results in S6 Table. Again, our coefficients of interest are still statistically significant. In terms of magnitude, they are smaller but within one standard deviation of our baseline results.

**Clustering.** In our main model, we cluster the standard errors at the county and year levels. A potential issue with this approach is that the asymptotics in clustering requires that both dimensions grow without limit [75,76], and the number of years in our setting is unlikely to be sufficient to satisfy this assumption. This paragraph shows that our results do not change if we cluster the standard error at a different level. More specifically, we report the results in S7 Table and show that our results do not change if we cluster standard errors at the county level, at the state level, and at the state and year levels.

**The election of Obama and the Great Recession.** In this paragraph, we consider that the Great Recession was a simultaneous event to the election of President Barack Obama in 2008. In order to exclude the hypothesis that this event affects our findings, we include in our regression a measure of the severity of the Great Recession. To do so, we follow [77] and quantify the severity of the Great Recession using unemployment changes between 2007 and

2009 in each county. We report the estimation results in S8 Table and show that our findings are not affected.

**Placebo tests.** To further test the validity of our identification strategy, we conduct a placebo test using non-voters and independent voters. Specifically, we consider the following two variables: i. the share of people in the county who did not vote during turnover Presidential elections, and ii. the share of votes for the independent party. These variables are standardized to facilitate comparison, allowing the interaction coefficient to represent the effect of a one-standard-deviation change on the outcome variable.

The results are presented in Columns (2) and (3) of S9 Table. Our findings indicate that turnover elections do not significantly impact the health of these individuals. Moreover, the coefficients are smaller compared to the coefficient associated with the standardized baseline effect reported in Column (1).

To provide further support for the validity of our parallel trend assumption, we conduct a placebo test based on the Presidential election of 2012. Since Obama won the election again, there has been no political turnover. Consequently, when we analyze this election event and focus our analysis on the period 2008-2015, we do not find any statistically significant effect, as reported in the last column of S9 Table. This suggests that our results are not driven by other simultaneous events related to Presidential elections or other confounding county dynamics.

**Alternative measures of individuals' health.** In this paragraph, we show that our results hold when we consider a self-reported measure of general health status from the BRDSS. More specifically, we follow previous literature (e.g., [52]) and rely on the following questions to measure the health of an individual: *"Would you say that in general, your health is excellent, very good, good, fair, poor?"*. We assigned a maximum value of 4 to "Excellent" and a minimum value of 0 to "Poor".

We estimate Eq 5 using as the outcome variable of interest the general health status and report the results in S10 Table. In line with our previous results, we find that general health gets worse for individuals located in Republican counties after the Obama election. More specifically, one standard deviation increase in the Republican vote share decreases the general health status by 0.6 percentage points.

**A long survey database.** Since our results are based on two turnover Presidential elections, in this paragraph, we provide further external validity to our results by considering a long survey database starting back in 1973 with information on individuals' general health status and their political preferences.

For this exercise, we estimate the following Eq (9):

$$Health_{j,t} = \beta Democrat_j \times President\ Democrat_t + \delta Democrat_j + \Gamma X_{j,t} + \theta_t + \omega_r + \epsilon_{j,t} \quad (9)$$

*Health* is the self-reported measure of general health of individual *j* at year *t*. *Democrat* is a dummy variable equal to one if an individual refers to himself as a strong Democrat. Our results do not change if we consider as democrat also individuals that report being "not a very strong democrat". *President Republican* is a dummy variable equal to 1 if the President is a Republican. We obtain a list with all the presidents of the United States and their political affiliation from Wikipedia at the following link: https://en.wikipedia.org/wiki/List_of_presidents_of_the_United_States . $X_{j,t}$ is a set of controls for the individuals' characteristics (dummy variables for each income category, a dummy variable equal to 1 if the individual is married, a dummy variable equal to 1 if the individual is a female, age, and age squared). $\omega_r$ and $\theta_t$ are respectively region and year fixed effects.

We show the estimation results in S11 Table. According to our preferred specification reported in the last column, being a Democrat when the President is also a Democrat improves the measure of self-rated health status by 3.4 percentage points. Considering that the average outcome value in this survey is 1.99, the coefficient of interest implies a decrease in health equal to 1.8 % with respect to the average.

**Additional control variables.** In an additional test, we demonstrate the robustness of our main results by incorporating the following additional control variables (interacted with the Post variable) into our main specification: a) the median age in the county, b) the percentage of the population with a BA or higher, c) the percentage of the population with health insurance in each county, and d) an indicator variable equal to one for minority counties. The detailed results are reported in S12 Table; in terms of magnitude, the coefficients are within one standard deviation of the baseline coefficients.

**Anticipation effects in Obama election?** In Fig 1, there is a noticeable dip in mortality trends for both Democratic and Republican counties between 2006 and 2007, before the election of Obama. This could reflect a broader national improvement in health outcomes affecting all counties (for example, due to aggregate policy changes or demographic shifts). However, it is also important to notice a more substantial decrease in the mortality rate in Democratic counties.

In order to test whether this difference is statistically significant and whether it leads to a violation of the parallel trends assumption, we re-estimated Eq (3) using 2006 as the reference year. The results are reported in S4 Fig. Importantly, we find that the interaction coefficient between the indicator variable for the year 2007 and the share of Democrats is not statistically significant.

**Internal and external causes of death.** In an additional exercise, we analyze the dynamic effects of internal and external causes of death — the two primary categories defining mortality. The results are presented in S5 Fig and S6 Fig. We still find that the assumption of parallel trends is likely to hold, since the coefficients are not statistically significant before the event. Additionally, our analysis shows that internal causes of death are the primary determinants of our results. On the other hand, the coefficients related to external causes of death are all close to zero.

**Alternative treatment measure.** To measure the exposure to the treatment of each county, we use a continuous variable based on the share of voters during the turnover Presidential elections. [78] recently showed that a difference-in-differences approach with a continuous treatment variable fails to have *causal* interpretations even when the treatment period is unique for all the observations. In fact, these estimators do not ascertain average causal response parameters under the standard parallel trend assumption; instead, they necessitate an alternative and more stringent assumption known as the *"strong parallel trends assumption"*. This assumption requires that the trajectory of outcomes for "lower-dose" units should mirror how outcomes for "higher-dose" units would have changed if they had received the "lower dose" instead. Without this condition, comparisons across dosage groups include causal responses but are contaminated by an additional term involving potentially different treatment effects of the same dose across different dosage groups—referred to as *selection bias*.

To deal with this potential concern, [78] suggest summarizing average level treatment effects among treated units by comparing the average change in outcomes for all treated units to the average change in outcomes for untreated units [79,80]. While in our setting completely "untreated" counties are not available since there is always a positive share of the population that voted for the losing President, we follow [18] and consider an alternative specification of

Eq (3) in which we propose as treatment an indicator variable equal to one if the share of voters during the turnover Presidential election is greater than the median value, and zero otherwise. Importantly, as reported in S7 Fig, we consistently find a negative effect on mortality after turnover Presidential elections and also show that the standard parallel trend assumption is likely to hold in this setting considering the absence of significant pre-event treatment coefficients.

It is important to note that while this exercise is suggestive and supports our causality claims, it does not guarantee that our estimators are unbiased. In fact, our new exercise does not ensure that the strong parallel trends assumption holds for every possible treatment level.

## 3 Conclusion

An unprecedented increase in political polarization has been documented in the United States and around the world during the last decades. Understanding how political sentiments and partisanship affect economic outcomes and community well-being is of crucial importance. Our paper contributes to this literature by providing evidence that a change in political leadership and political attitudes can affect health outcomes.

Our robust results show that mortality rates increased in counties supporting the losing Presidential candidate after the turnover Presidential elections. More specifically, we find an average increase in mortality in losing counties equal to 7 individuals per 100'000 population. The effect is economically meaningful, immediate, and persistent. Also, we provide consistent results using alternative databases and spanning periods.

Evidence suggests that political sentiments and social isolation may be important factors underlying our findings. Indeed, we document after turnover Presidential elections a decrease in the degree of social interactions in the losing counties. Also, we identify a further increase in sudden causes of death about crucial political events and a worsening of the mental health of the individuals. Unfortunately, we do not empirically assess their relative contribution to the observed effects on mortality, and we cannot exclude the possibility that other mechanisms also play a role. However, on the other side, we do not find any evidence that political partisanship affects economic prosperity or that partisan groups benefit from the economic policies of the governments from an economic point of view.

Our findings have important policy implications. These show indeed that the lack of political representation and political sentiments lead to high costs in terms of health and households' welfare, even without any change in local economic prosperity. Furthermore, the rise in political polarization is likely to further increase the economic costs estimated in our analysis. Our findings suggest that ensuring political representation and introducing policies aimed at building trust in government and improving public perceptions of political institutions could be beneficial for the health of the communities.

## Supporting information

**S1 Text. Online Appendix.**
(PDF)

**S1 Fig. Sensitivity analysis to violations of parallel trends.**
(PDF)

**S2 Fig. Obama election and Black mortality.**
(PDF)

**S3 Fig. Added-variable (partial residual) plots.**
(PDF)

**S4 Fig. Event study with 2006 as reference year.**
(PDF)

**S5 Fig. Internal causes of death: event-study effects.**
(PDF)

**S6 Fig. External causes of death: event-study effects.**
(PDF)

**S7 Fig. Alternative treatment (median split) event study.**
(PDF)

**S1 Table. Population weights for age-adjustment.**
(PDF)

**S2 Table. Variable definitions and data sources.**
(PDF)

**S3 Table. Mental health robustness at the state level.**
(PDF)

**S4 Table. Normalized differences: losing vs. winning counties.**
(PDF)

**S5 Table. Obama and Black mortality (DiD at county–race–year level).**
(PDF)

**S6 Table. Outlier robustness (winsorization).**
(PDF)

**S7 Table. Alternative clustering of standard errors.**
(PDF)

**S8 Table. Controlling for Great Recession severity.**
(PDF)

**S9 Table. Placebo tests and alternative political groups.**
(PDF)

**S10 Table. General health (BRFSS) outcomes.**
(PDF)

**S11 Table. Long-run survey evidence (GSS, 1973–present).**
(PDF)

**S12 Table. Additional controls.**
(PDF)

## Author contributions

**Conceptualization:** Sris Chatterjee, Iftekhar Hasan.

**Data curation:** Stefano Manfredonia.

**Formal analysis:** Stefano Manfredonia.

**Methodology:** Stefano Manfredonia.

**Project administration:** Sris Chatterjee.

**Software:** Stefano Manfredonia.

**Writing – original draft:** Sris Chatterjee, Iftekhar Hasan.

**Writing – review & editing:** Sris Chatterjee, Iftekhar Hasan.

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
