## [Decision Letter · Decision Letter 0]

12 Feb 2025

PONE-D-24-56123Partisan HealthPLOS ONE

Dear Dr. Manfredonia,

Thank you for submitting your manuscript to PLOS ONE. After careful consideration, we feel that it has merit but does not fully meet PLOS ONE’s publication criteria as it currently stands. Therefore, we invite you to submit a revised version of the manuscript that addresses the points raised during the review process.

I have received a positive review of one expert reviewer, which aligns with my personal view of the manuscript. Unfortunately, a second reviewer did not submit their evaluation on time. To expedite the peer-review process, I would like to invite you to revise your manuscript based on the first reviewer's comments. However, please be aware that I will look for a second reviewer to evaluate the revised manuscript, too. Reviewer 1 makes a number of important comments regarding the empirics and presentation of the data. All of these seem highly relevant. However, I would like to stress that the issue raised regarding the argumentation should also be given sufficient attention. A carefully executed empirical analysis is only of value if we understand that the results are plausible and how we can make sense of them. And for that a solid theoretical foundation is needed.

We look forward to receiving your revised manuscript.

Kind regards,

Jerg Gutmann

Academic Editor

PLOS ONE

Journal Requirements:

2. Since you are reporting a retrospective study of medical records or archived samples, please ensure that you have discussed whether all data were fully anonymized before you accessed them and/or whether the IRB or ethics committee waived the requirement for informed consent. If patients provided informed written consent to have data from their medical records used in research, please include this information.

Reviewers' comments:

Reviewer's Responses to Questions

**Comments to the Author**

1. Is the manuscript technically sound, and do the data support the conclusions?

Reviewer #1: Yes

2. Has the statistical analysis been performed appropriately and rigorously? 

Reviewer #1: Yes

3. Have the authors made all data underlying the findings in their manuscript fully available?

Reviewer #1: Yes

4. Is the manuscript presented in an intelligible fashion and written in standard English?

Reviewer #1: Yes

5. Review Comments to the Author

Reviewer #1: Review of” Partisan Health” PONE-D-24-56123

The paper examines whether a change in political leadership impacts health outcomes. Specifically, the researchers analyze the impact of elections where power shifts, causing individuals to either align with or lose alignment with the party of the President. The findings indicate that a lack of political alignment has an immediate and long-lasting negative effect on health. The study finds no evidence that these effects can be explained by other confounding trends, economic changes, or policy shifts. Furthermore, the results suggest that political sentiments and social isolation play a role, and that the lack of political representation affects individuals' mental health.

The main question is relevant, and I find the manuscript very well written, but the paper should better explain how the current research contributes to the literature that is closest to the analysis and the research question. Also, I believe the authors can do more and discuss more in relation to key assumptions and recent developments of the DiD literature. Taken together my recommendation is to allow the authors to revise the paper for possible publication in PLOS One.

COMMENTS

CONTRIBUTION

I think the overview of the literature is ok but would prefer that the authors are more to the point. It seems to me there are some existing works that are mentioned in the manuscript that relate very closely to the work in this study, and they should be elaborated more upon and how this piece contributes more in detail in relation to these papers.

IDENTIFICTAION AND POTENTIAL THREATS

(a) PARALELL TREND ASSUMPTION

The authors show graphically that there are reasons to believe that the parallel trend assumption holds for overall mortality. I think it is important to also who this for specific death causes as this is also a focus in the analysis. Similarly, I am not sure the authors present graphs for specific groups examined.

Also, I think it is relevant to discuss the likeliness of the parallel trend assumption to hold in relation to previous academic work showing that there are health effects before elections/shifts in power.

And finally, in Figure 1, what is it that we see in figure (a) between 2006 and 2007?

(b) IDENTIFICATION

In my view the causal claims should be restricted to the analysis regarding mortality.

It is good that the authors try to take recent developments of the DiD method into account, but I would prefer more of elaboration regarding the potential identification problem that they have in the main specification/results and what is not fully solved by the suggested solution by Dahl et al (2022).

ARGUMENTATION

The hypothesis to why political sentiments as a mechanism would correlate with sudden deaths is not well motivated. I think the argumentation must be strengthened and related to previous literature on this point.

DATA

Is it noted in the death certificate data if a person has a different nationality or not recorded in a census? I think there are reasons to believe that e.g. that people in the US illegally would be more stressed when there are political shifts of different kinds as the ones examined here. Is there a possibility to sort out the death rates among non-US citizens dying in the counties?

I am not very fond of the use of the membership association data. Rather than the number of organizations it should focus on the number of people being members in associations. I am sure this is reported in the NETS database.

Is the BRFSS data representative at the county level?

I believe the restricted GSS data do include information on the country of residence. Not that I suggest you should use it but rather mention it if I recall things correctly.

Please mention the number of counties in the US in the data section.

6. PLOS authors have the option to publish the peer review history of their article (what does this mean?). If published, this will include your full peer review and any attached files.

Reviewer #1: No

---

## [Author Response · Author response to Decision Letter 1]

31 Mar 2025

* We updated a PDF file with our reply to the referee. Indeed, we could not update the figures here. *

Revision of the manuscript “Partisan Health”

Dear Referee,

We would like to thank you very much for your thoughtful and constructive feedback, which helped us to improve the paper substantially.

Before addressing the specific points you raised, we would like to present an overview of the significant modifications made in the revised version. More specifically, the key changes can be summarized as follows:

• We have explained more clearly how our paper contributes to the existing literature, and we have provided a more detailed comparison between our paper and related existing studies.

• We discuss in more detail and test the parallel trend assumption and its validity in the context of our paper.

• We explain more clearly the theoretical background behind our findings.

In the remainder, we put your comments and suggestions in italics, followed by our

detailed response.

With best regards,

The authors

Comments to the author

The paper examines whether a change in political leadership impacts health outcomes. Specifically, the researchers analyze the impact of elections where power shifts, causing individuals to either align with or lose alignment with the party of the President. The findings indicate that a lack of political alignment has an immediate and long-lasting neg- ative effect on health. The study finds no evidence that these effects can be explained by other confounding trends, economic changes, or policy shifts. Furthermore, the results suggest that political sentiments and social isolation play a role, and that the lack of po- litical representation affects individuals’ mental health.

The main question is relevant, and I find the manuscript very well written, but the paper should better explain how the current research contributes to the literature that is closest to the analysis and the research question. Also, I believe the authors can do more and discuss more in relation to key assumptions and recent developments of the DiD lit- erature. Taken together my recommendation is to allow the authors to revise the paper for possible publication in PLOS One.

Reply: Thank you very much for your assessment of our paper. We hope you find the revised version clearer regarding its contribution to the literature and how it differs from previous studies. Additionally, we hope you find the expanded discussions on the key as- sumptions behind the validity of our empirical approach, as well as the additional results supporting its validity, to be adequate.

COMMENTS

CONTRIBUTION

I think the overview of the literature is ok but would prefer that the authors are more to the point. It seems to me there are some existing works that are mentioned in the manuscript that relate very closely to the work in this study, and they should be elaborated more upon and how this piece contributes more in detail in relation to these papers.

Reply: Thank you for your comment. In the revised submission, we have provided a clearer explanation of how our paper differs from the cited works in the final paragraph of the introduction. Specifically, we have added the following comments.

Maas and Lu (2021) identified a positive correlation between mortality rates and parti- san losses. In a similar vein, Wasfy et al. (2017) and Goldman et al. (2019) use regression analysis to demonstrate a correlation between health metrics and voting patterns, with voting outcomes serving as the dependent variable. Our study differs from these works by providing causal, unidirectional estimates of the impact of a shift in political power on mortality, leveraging turnover elections as natural experiments. In addition to this, our unique county-race-year level analysis of black mortality following Obama’s election provides a within-county comparison that controls for unobserved county characteristics, which is a methodological improvement over studies that rely solely on cross-county com- parisons. More importantly, we investigate the specific mechanisms driving these effects, emphasizing the role of political sentiments and social isolation while ruling out changes in economic characteristics as a potential channel. Overall, our analysis offers a more comprehensive and precise understanding of the relationship between health outcomes

and political leadership.

Pierce et al. (2016) show that elections affected the happiness of partisan losers, but this effect dissipated within a week. Our study demonstrates that health effects are not transient but persist over years, suggesting a different and more concerning mechanism than temporary emotional responses. Furthermore, our long-term GSS analysis from 1973 provides historical context missing from most contemporary studies, offering evidence that the health-politics relationship has been consistent across multiple administrations.

While Mian et al. (2021) and Barsky and Sims (2012) focused on economic senti- ments and consumption following political leadership changes, our paper examines both economic and non-economic pathways, showing that social isolation and mental health deterioration drive effects even in the absence of economic changes. We test for both sudden causes of death and chronic conditions, providing evidence for both immediate stress responses and cumulative effects, which is a more comprehensive approach than studies focusing on single pathways.

Our paper directly addresses the contradictory findings of Classen and Dunn (2010) by suggesting that the benefits of social integration they observed in 1981-2005 may have been outweighed by increasing polarization in your more recent study period. We connect seemingly disparate literature streams (political polarization, social capital, and public health) in a coherent framework that helps explain why some previous studies found effects while others did not.

Unlike studies focusing on specific health behaviors (like Milosh et al. (2021) on mask- wearing), our paper quantifies the overall mortality burden of political misalignment, making it more relevant for broad public health policy. Our calculation of economic costs using FEMA’s value of statistical life provides a concrete policy-relevant metric absent from most previous research on partisan health differences.

IDENTIFICTAION AND POTENTIAL THREATS

(a) PARALELL TREND ASSUMPTION

The authors show graphically that there are reasons to believe that the parallel trend as- sumption holds for overall mortality. I think it is important to also who this for specific death causes as this is also a focus in the analysis. Similarly, I am not sure the authors present graphs for specific groups examined.

Reply: We thank you for this input. In the revised version of the paper, we present graphs supporting the parallel trend assumption for the following outcome variables and events:

• Overall mortality rates in Democratic and Republican counties after the turnover presidential elections of Presidents Trump and Obama, respectively.

• Black morality rates after the turnover election of President Obama.

• Quarterly mortality rates for sudden causes of death for the turnover presidential

elections of Presidents Trump and Obama, respectively.

We report all the graphs in Figures (Referee 1-1)-(Referee 1-4), as well as in the revised version of the paper. More specifically, these results show that before a turnover in the Presidential elections, mortality rates are similar between Democratic and Republican counties, but they diverge afterward depending on the outcome of the election.

In an additional exercise reported in a paragraph in sub-section 2.4 in the revised version of the paper, we analyze the long-term dynamic effects of internal and external causes of death—the two primary broad categories defining mortality. The results are presented in Figures (Referee 1-5) and (Referee 1-6). We still find that the assumption of parallel trends is likely to hold since none of the coefficients is significant before the treatment. Additionally, our analysis shows that internal causes of death are the primary determinants of our results. On the other hand, the coefficients related to external causes of death are all close to zero even after turnover elections.

Also, I think it is relevant to discuss the likeliness of the parallel trend assumption to hold in relation to previous academic work showing that there are health effects before elections/shifts in power.

Reply: Thank you for this point. You raise an important point about potential pre- election health effects. Indeed, our analysis shows increased mortality in politically polar- ized counties before Presidential elections, consistent with Panagopoulos and Weinschenk (2022) who document deteriorating mental health during campaign periods. However, we discuss that these pre-election effects do not invalidate our difference-in-differences approach for two reasons:

• First, our time series graphs show that the mortality trends in Democratic and Republican counties remain parallel prior to the election. We provide a more detailed discussion of Figure (1) in our paper in response to your next question.

• Second, while campaign stress affects all counties, its elevation is likely to be similar across partisan counties before the election, with divergence only occurring after the election outcome is known.

We have enhanced our discussion in sub-section 2.2 in the revised version of the paper to address these considerations explicitly.

And finally, in Figure 1, what is it that we see in figure (a) between 2006 and 2007?

Reply: Thank you for this remark. In Figure 1(a) in the paper, there is a noticeable dip in mortality trends for both Democratic and Republican counties between 2006 and 2007, before the election of Obama. This could reflect a broader national improvement in health outcomes affecting all counties (for example, due to aggregate policy changes or demographic shifts). However, it is also important to note that a more substantial decrease in the mortality rate occurs in Democratic counties.

To test whether this difference is statistically significant and whether it leads to a vio- lation of the parallel trends assumption, we re-estimated Equation (3) in the paper using 2006 as the reference year. The results are reported in Figure (Referee 1-7). We find that the interaction coefficient between the indicator variable for the year 2007 and the share of Democrats is not statistically significant and close to zero. Therefore, the greater decrease in mortality in Democratic counties before the election is unlikely to affect our findings or violate the assumption of parallel trends. We have commented on this result in a paragraph in sub-section 2.4 of the revised paper.

(b) IDENTIFICATION

In my view the causal claims should be restricted to the analysis regarding mortality. It is good that the authors try to take recent developments of the DiD method into account, but I would prefer more of elaboration regarding the potential identification problem that they have in the main specification/results and what is not fully solved by the suggested solution by Dahl et al (2022).

Reply: We agree that our causal claims should be focused on the mortality analysis, where our identification strategy is most robust. In the revised manuscript, we have clar- ified the distinction between our causal findings on mortality and the more suggestive evidence regarding mechanisms. We have adjusted our language accordingly through- out to ensure we do not overstate causal interpretations beyond what our identification strategy supports.

Regarding your concern about the difference-in-differences implementation, we have expanded our discussion of potential identification challenges in Section 5. In addition, while we follow Dahl et al. (2022) in using threshold-based binary treatment assignment as a robustness check, we acknowledge this does not fully address all identification con- cerns with continuous treatment variables highlighted in recent econometric literature (Callaway et al., 2024). More specifically, we explicitly discussed the limitations of our continuous treatment approach and clarified that, although our new exercise is suggestive and supports our causality claims of an effect of turnover elections on the health of the individuals, it does not guarantee that our estimators are unbiased. In fact, our new exer- cise does not ensure that the strong parallel trends assumption holds for every treatment level.

We believe these additions in a paragraph in sub-section 2.4 in the revised version of the paper substantially strengthen our paper by being transparent about identification challenges while demonstrating that our core findings on mortality effects remain robust across alternative specifications and sensitivity analyses. Thank you again for pushing us to improve the rigor of our analysis.

ARGUMENTATION

The hypothesis to why political sentiments as a mechanism would correlate with sudden deaths is not well motivated. I think the argumentation must be strengthened and related to previous literature on this point.

Reply: Thank you for this important observation. We acknowledge that our explana- tion of why political sentiments would correlate with sudden deaths requires stronger motivation and connection to established literature. In the revised manuscript, we have substantially expanded subsection 2.3.1 with the following comments.

First, we have strengthened the theoretical foundation by drawing on the extensive stress-health literature. Acute psychological stressors trigger well-documented physio- logical responses, including increased cardiovascular reactivity, inflammatory responses, and dysregulation of the hypothalamic-pituitary-adrenal axis (McEwen, 1998; Kivimäki and Steptoe, 2018). These mechanisms provide biological pathways connecting sudden political disappointment to acute health events.

Second, we have incorporated research specifically on political events and acute health outcomes. We cite Rosman et al. (2021), who documented increased arrhythmia risk dur- ing the 2016 presidential election, and Mefford et al. (2020), who found elevated cardio- vascular hospitalizations around the same period.

Third, we connect our findings to the concept of ”broken heart syndrome” (takotsubo cardiomyopathy), where sudden emotional stress can trigger cardiovascular events even in otherwise healthy individuals (Templin et al., 2015). This literature provides a medical mechanism for how election-related distress could manifest as sudden mortality.

With these additions, we believe our argument for political sentiments as a mecha- nism for sudden deaths is now much better grounded in both theoretical frameworks and empirical evidence from multiple disciplines.

DATA

Is it noted in the death certificate data if a person has a different nationality or not recorded in a census? I think there are reasons to believe that e.g. that people in the US illegally would be more stressed when there are political shifts of different kinds as the ones examined here. Is there a possibility to sort out the death rates among non-US citizens dying in the counties?

Reply: Thank you for raising your question regarding nationality in our mortality data. In the US, death certificates record an individual’s residence status (i.e., whether the person resides outside the U.S.) and the decedent’s place of birth, which can indicate a foreign birth.

While your point about differential stress among non-citizens during political transitions is well-founded and relevant, there are significant limitations to addressing this directly with our data. For what concerns the residence information, only 0.1% of the deceased individuals have a residence in a foreign country. On the other hand, regard- ing the birthplace information, a potential problem with this variable is that it does not distinguish between legal permanent residents, naturalized citizens, visa holders, or undocumented immigrants for people that have born outs

---

## [Decision Letter · Decision Letter 1]

12 Aug 2025

PONE-D-24-56123R1Partisan HealthPLOS ONE

Dear Dr. Manfredonia,

Thank you for submitting your manuscript to PLOS ONE. After careful consideration, we feel that it has merit but does not fully meet PLOS ONE’s publication criteria as it currently stands. Therefore, we invite you to submit a revised version of the manuscript.

As indicated in my invitation to revise and resubmit, a second reviewer had to evaluate the revised version of the manuscript. Unfortunately, several reviewers first agreed to review and then did not deliver a report on time. I have now received a second report and both reviewers agree that the revised paper can be accepted for publication. However, before we can accept your article, I must ask you to implement minor revisions to bring the manuscript in line with the PLOS ONE publication criteria: - Please modify the title to ensure that it is meeting PLOS' guidelines (https://journals.plos.org/plosone/s/submission-guidelines#loc-title). In particular, the title should be "specific, descriptive, concise, and comprehensible to readers outside the field". In this case, the title is not sufficiently informative and specific about your study's scope and methodology. Adding a descriptive subtitle would be one way to satisfy these requirements.- You empirical analysis is exploring potential causal mechanisms, but it is missing an explicit statement as to the limitations of this analysis. Some statements in the manuscript might, thus, appear as not being fully supported by the evidence provided. On the one hand, some of these mechanisms, such as social isolation and mental health are not necessarily independent from each other. On the other hand, it cannot be ruled out that important other confounding mechanisms are not accounted for. Different mechanisms are not (even where it would be theoretically plausible) tested against each other. Thus, I would ask you to address these limitations, e.g., in the following way:(i) At the beginning of Section 2.3, clarify that you are highlighting <potential> mechanisms behind your findings.(ii) State explicitly in the manuscript that you only show that these potential mechanisms exist, but you are not empirically evaluating their relative importance for mortality by testing them against each other.(iii) Finally, if you agree that the analysis of mechanisms has these limitations, please tone down the sentence in the conclusion stating that "We further show that political sentiments and social isolation are at the core of our findings." toward something like "these effects appear to be due to ..." or "evidence suggests that ...".===========================

A rebuttal letter that responds to the requested revisions and states if and how they have been implemented.A marked-up copy of your manuscript that highlights changes made to the original version. You should upload this as a separate file labeled 'Revised Manuscript with Track Changes'.An unmarked version of your revised paper without tracked changes. You should upload this as a separate file labeled 'Manuscript'.

We look forward to receiving your revised manuscript.

Kind regards,

Jerg Gutmann

Academic Editor

PLOS ONE

Journal Requirements:

Reviewers' comments:

Reviewer's Responses to Questions

**Comments to the Author**

1. If the authors have adequately addressed your comments raised in a previous round of review and you feel that this manuscript is now acceptable for publication, you may indicate that here to bypass the “Comments to the Author” section, enter your conflict of interest statement in the “Confidential to Editor” section, and submit your "Accept" recommendation.

Reviewer #1: All comments have been addressed

Reviewer #2: All comments have been addressed

2. Is the manuscript technically sound, and do the data support the conclusions?

Reviewer #1: Yes

Reviewer #2: Yes

3. Has the statistical analysis been performed appropriately and rigorously? 

Reviewer #1: Yes

Reviewer #2: Yes

4. Have the authors made all data underlying the findings in their manuscript fully available?

Reviewer #1: No

Reviewer #2: No

5. Is the manuscript presented in an intelligible fashion and written in standard English?

Reviewer #1: Yes

Reviewer #2: Yes

6. Review Comments to the Author

Reviewer #1: I am happy with the revised version of the paper and that the authors tried to handle the comments raised in the best possible ways.

Reviewer #2: (No Response)

7. PLOS authors have the option to publish the peer review history of their article (what does this mean?). If published, this will include your full peer review and any attached files.

Reviewer #1: No

Reviewer #2: No

---

## [Author Response · Author response to Decision Letter 2]

19 Aug 2025

Revision of the manuscript PONE-D-24-56123R1

Dear Professor Gutmann,

We want to thank you and the reviewers for your feedback. We have carefully reviewed the manuscript to address the points raised. In the following, we provide a detailed, point by-point response.

Comment 1: Please modify the title to ensure that it is meeting PLOS’ guidelines

(https://journals.plos.org/plosone/s/submission-guidelines#loc-title). In particular, the title should be ”specific, descriptive, concise, and comprehensible to readers outside the field”. In this case, the title is not sufficiently informative and specific about

your study’s scope and methodology. Adding a descriptive subtitle would be one way to satisfy these requirements.

Response: Thank you for your guidance. We have changed the title to:

“The Health Costs of Losing Political Representation: Evidence from U.S. Presidential Elections”

This title clearly conveys the scope (health impacts), the key independent variable (loss of political representation), and the empirical context (U.S. Presidential elections). For this reason, we believe it meets PLOS’ requirements for specificity, descriptiveness, conciseness,

and accessibility to non-specialists.

Comment 2: You empirical analysis is exploring potential causal mechanisms, but it is missing an explicit statement as to the limitations of this analysis. Some statements in the manuscript might, thus, appear as not being fully supported by the evidence provided. On

the one hand, some of these mechanisms, such as social isolation and mental health are not necessarily independent from each other. On the other hand, it cannot be ruled out that important other confounding mechanisms are not accounted for. Different mechanisms

are not (even where it would be theoretically plausible) tested against each other. Thus, I would ask you to address these limitations, e.g., in the following way:

(i) At the beginning of Section 2.3, clarify that you are highlighting potential mechanisms behind your findings.

(ii) State explicitly in the manuscript that you only show that these potential mechanisms exist, but you are not empirically evaluating their relative importance for mortality by testing them against each other.

(iii) Finally, if you agree that the analysis of mechanisms has these limitations, please tone down the sentence in the conclusion stating that ”We further show that political sentiments and social isolation are at the core of our findings.” toward something like

”these effects appear to be due to ...” or ”evidence suggests that ...”.

Response: Thank you again for your guidance. We have implemented the requested clarifications:

1. At the start of Section 2.3, we now state: “The previous section provides evidence that a change in political leadership affects health outcomes. We recognize that multiple factors could simultaneously contribute to this result. Furthermore, our reduced form analysis does not provide us with the capability to disentangle the extent to which each mechanism we investigate can explain the documented results. However, in this section, we still highlight interesting potential underlying mechanisms behind our main finding. ” We also changed our wording throughout the whole paper accordingly.

2. We follow your suggestion, and we added this sentence to the conclusions: “Unfortunately, we do not empirically assess their relative contribution to the observed effects on mortality, and we cannot exclude the possibility that other mechanisms

also play a role.” We further include similar statements in the introduction and throughout the paper.

3. We toned down the original conclusion sentence. It now reads: “Evidence suggests that political sentiments and social isolation may be important factors underlying our findings.”

We hope that these changes ensure that that our discussion of mechanisms is appropriately framed as exploratory and that our claims are aligned with the evidence presented.

We believe these revisions address the requested changes. We thank you again for your helpful comments and guidance and look forward to the continued evaluation of our manuscript.

Sincerely,

Sris Chatterjee, Iftekhar Hasan, Stefano Manfredonia

---

## [Editor Report · Decision Letter 2]

30 Sep 2025

The Health Costs of Losing Political Representation: Evidence from U.S. Presidential Elections

PONE-D-24-56123R2

Dear Dr. Manfredonia,

We’re pleased to inform you that your manuscript has been judged scientifically suitable for publication and will be formally accepted for publication once it meets all outstanding technical requirements.

Kind regards,

Jerg Gutmann

Academic Editor

PLOS ONE
---

## [Editor Report · Acceptance letter]

PONE-D-24-56123R2

PLOS ONE

Dear Dr. Manfredonia,

I'm pleased to inform you that your manuscript has been deemed suitable for publication in PLOS ONE. Congratulations! Your manuscript is now being handed over to our production team.

Kind regards,

on behalf of

Prof. Dr. Jerg Gutmann

%CORR_ED_EDITOR_ROLE%

PLOS ONE